# SuCCESs – a global IAM for exploring the interactions between energy, materials, land-use and climate systems in long-term scenarios (model version 2024-10-23)

Tommi Ekholm[1], Nadine-Cyra Freistetter[1], Tuukka Mattlar[1], Theresa Schaber[1], Aapo Rautiainen[1,2]

[1]Climate System Modelling, Finnish Meteorological Institute, P.O. BOX 503, 00101 Helsinki, Finland
[2]Environmental and Natural Resource Economics, Natural Resources Institute Finland (Luke), Latokartanonkaari 9, 00790 Helsinki, Finland

*Correspondence to*: Tommi Ekholm (tommi.ekholm@fmi.fi)

**Abstract.** SuCCESs is a bottom-up Integrated Assessment Model (IAM) that represents energy production and use, materials production, land-use and climate globally. The primary use-case for SuCCESs is to calculate long-term scenarios until 2100 that consider the interactions between these systems, for example the greenhouse gas emissions from energy, materials and land-use and their impact on climate change. The four systems are hard-linked in SuCCESs and scenarios are solved through intertemporal optimization by minimizing discounted system costs to satisfy projected demand and other

constraints, e.g. climate targets. This yields a long-term equilibrium solution between the modelled systems. This article introduces the model logic and structure, describes the overall representation of each system, and provides an evaluation by comparing the scenarios produced by SuCCESs with different end-of-century radiative forcing targets to those from other IAMs. Towards this end, and to demonstrate the capability of SuCCESs for large-scale scenario exploration, we also conduct a sensitivity analysis employing Monte Carlo sampling with a 1000-member scenario ensemble for each radiative forcing

target. Last, we discuss some practical aspects and different ways of using the model in long-term scenario analyses.

## 1   Introduction

SuCCESs (Sustainable Climate Change mitigation strategies in Energy-land-material Systems) [1] is a lightweight, technologically detailed, global Integrated Assessment Model (IAM) that focuses on representing the interactions between energy, materials, land-use and climate systems in long-term scenarios. This article introduces the model's overall structure,

describes the representation of each system, and demonstrates its use through a set of mitigation scenarios reaching the RCP radiating forcing targets by 2100 (van Vuuren et al., 2011), and a sensitivity analysis employing Monte Carlo sampling.

---

[1] Model version used in this manuscript: 2024-10-23

IAMs are an essential tool in modelling long-term strategies to mitigate climate change. They are often categorized broadly into two types: bottom-up, process-based IAMs and top-down, cost-benefit IAMs (Keppo et al., 2021; Weyant, 2017). Both portray greenhouse gas (GHG) emission pathways, their connection to societal and economic functions, and the climate change induced by GHG emissions over the long-term, often until 2100 and sometimes beyond. The main differences between the model types are the driving force for emission reductions and the detail in representing from what activities emissions arise and how they can be reduced. Top-down IAMs present economic activity in an aggregated manner and seek to calculate economically optimal mitigation pathways by balancing marginal costs and benefits from mitigation, often without considering emission sources or technological options for reducing emissions explicitly. For example, DICE (Nordhaus, 1992, 2017) – the best-known top-down IAM – considers the aggregate global economy and uses stylized mathematical formulas to estimate the impact of climate change and mitigation action on economic output.

In contrast, bottom-up, process-based IAMs explicitly represent the economic activities that cause GHG emissions, such as energy use, industrial production, and transportation. These models also portray explicitly the current and future technological options to reduce emissions. They typically require a prescribed climate policy – an emission limit or tax, or a temperature target, for example – and model in more detail how these policies affect the activities that produce or reduce emissions.

SuCCESs is a bottom-up, process-based IAM. It models explicitly the investments in, and the operation of technological processes, and tracks the flows of commodities and emissions to and from these processes. Prominent process-based IAMs differ from each other in terms of their solution method, model scope, and the level detail towards technological, geographic and policy options (Keppo et al., 2021). In terms of solution concept and model structure, SuCCESs shares similarities with the MESSAGEix (Huppmann et al., 2019), TIAM (Loulou and Labriet, 2008) and GLUCOSE (Beltramo et al., 2021) IAMs. Yet, compared to the pool of existing process-based, global IAMs (and combinations of energy, land-use and climate models), SuCCESs provides a unique combination of using an intertemporal optimization solution method with hard-linked, bottom-up modelling of the energy, land-use and climate system. SuCCESs also differs from these models in terms of its single-region representation, which was chosen to maintain simplicity and low computational burden. Additionally, SuCCESs represents the production of main bulk materials and their linkage to energy use, land-use and climate. Similar expansion of model scope has also been pursued in other IAMs recently (Stegmann et al., 2022; Ünlü et al., 2024). Our ongoing model development aims to extend the representation of materials to cover the in-use stock of materials in the global economy. Given these features, SuCCESs is particularly suited to address research questions where the long-term development and interactions between the considered domains are important. As an example, we investigated the climate change mitigation potential of bioplastics in one of our first applications of the model – an issue with strong links between materials production, energy, land-use and climate change (Mattlar and Ekholm, 2025).

Given the computational ease of running the model, SuCCESs is particularly suitable for large-scale scenario exploration and stochastic applications. To demonstrate this, we conduct a parametric sensitivity analysis of the model using Monte

Carlo sampling, an exercise similar to (Panos et al., 2023). Also leveraging on the low computational burden, the model could be supplemented with capabilities for stochastic programming in the future, which would enable to find strategies that hedge against modelled uncertainties (Ekholm, 2014; Loulou and Kanudia, 1999).

This paper gives an overview of the model logic, its general structure and main features. As the model's components and numerical parameters are subject to updates in the future, this paper focuses on the overall structure of the model and how

the subsystems interact, as well as how the model can be used and what kinds of results a user can expect.

Numerical values are provided only in cases where we do not foresee an update to the values to be probable. Many of the parameter values are nevertheless specific to some scenario assumptions and are thus likely to be modified by the model user for the case-specific needs of each scenario experiment conducted with the model.

The paper is structured in the following way. Section 2 describes the structure and main features of the model. Section 3

presents a demonstration of the model, first through four scenarios reaching the RCP radiating forcing targets by 2100 (van Vuuren et al., 2011) and their comparison to the SSP scenario ensemble (Riahi et al., 2017), then through a sensitivity analysis by running a 1000 member scenario ensemble for each radiative forcing target with Monte Carlo sampling of main input parameters. Further, in section 4, we provide some practical guidance and viewpoints for model use. Finally, section 5 presents our concluding thoughts and discusses some future directions for model development and use.

**2    Model structure**

SuCCESs is a global, demand-driven partial equilibrium model that is solved through intertemporal optimization (linear programming) assuming perfect foresight. The objective is to minimize discounted system costs while satisfying exogenously set projections of inelastic commodity demands, adhering to structural equations, and meeting other optionally set constraints, such as climate targets. This solution corresponds with a long-term economic equilibrium with a projection of

inelastic demands for goods and services under assumptions of perfect markets and perfect foresight. Trade is not considered explicitly, due to the single-region implementation. With these assumptions, competition between producers leads to a least-cost solution to fulfil the demand, with the supply-side elasticity modelled bottom-up through technologies and resources with different costs; concomitantly maximizing the producer surplus (see e.g. Loulou and Labriet, 2008). A common use case of SuCCESs is to calculate long-term climate change mitigation scenarios, which can be incentivized through climate

targets or emission pricing. In both cases, SuCCESs seeks mitigation measures equally from energy, material production and land-use, considering the dynamics and interactions between these systems.

The model contains hard-linked modules representing the energy system, land-use, materials and climate. Key interactions between the modules are the energy, fossil feedstock and biomass requirements for producing materials, bioenergy production, and the GHG emissions and sinks from energy production and use, material production and land-use, including a

full accounting of terrestrial carbon stocks in vegetation and soil. In the current version of the model, the demand for

materials is specified exogenously, and is thus not affected by e.g. investments into energy production. Figure 1 illustrates the overall model structure and main interactions.

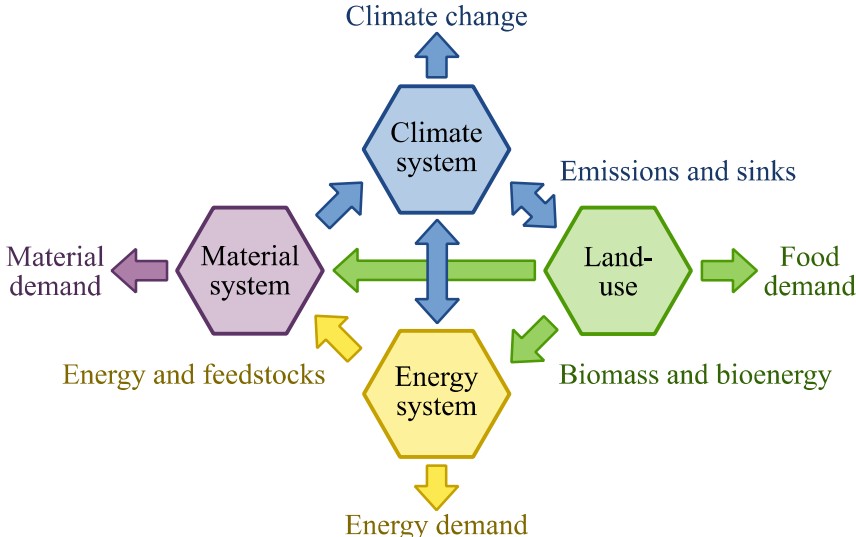

**Figure 1: An overview of the structure of the SuCCESs IAM, it's modules and their main interactions.**

The model covers the world as a single region, although land-use is represented with ten geographical biomes, the production and emissions from which are aggregated to global level. The base time unit for model variables is annual (e.g. energy flows are represented in PJ per year), but the model is solved at a decadal resolution (i.e., a single year is representative for a 10-year time period) until 2100. However, an hourly sub-module is used to represent the variations in wind and solar electricity production and exogenously varying electricity demand.

The main variables of the model comprise commodity flows, investments and operation of production processes, land area allocation for different uses, terrestrial carbon stocks, GHG emissions, and climate variables, like radiative forcing and global mean temperature change. All variables represent physical quantities, such as energy flow or production capacity (in PJ per year for energy and Mt per year for materials), land area (million km²) or carbon stock (GtC). Economics are accounted for through costs for these physical quantities, such as activity cost for running a process or investments costs for

installing new production capacity.

    A simplified problem statement of SuCCESs is given below. In the notation, lowercase letters refer to model input parameters and uppercase letters to model variables.

$$\min \sum_{t} (1+\beta)^{-t} \left[ \sum_{p} \left( c^{K}_{p,t}\, K_{p,t} + c^{A}_{p,t}\, A_{p,t} \right) + \sum_{e} c^{E}_{e,t}\, E_{e,t} \right] \tag{1}$$

so that

$$I_{p,c,t} = i_{p,c,t}\, A_{p,t}$$

$$O_{p,c,t} = o_{p,c,t}\, A_{p,t}$$

$$\sum_p O_{p,c,t} \geq \sum_p I_{p,c,t} + d_{c,t} \quad \forall c, t$$

$$A_{p,t} \leq f_{p,t}\, C_{p,t} \quad \forall p, t$$

$$C_{p,t} = \sum_{t-\tau_p \leq \tilde{t} \leq t} K_{p,\tilde{t}} \quad \forall p, t$$

$$E_{e,t} = \sum_p e_{e,p,t}\, A_{p,t} \quad \forall e, t,$$

and for land-use (in considerably simplified form):

$$\sum_u R_{b,u,t} = R_b \quad \forall t,$$

$$C_{\tilde{p},t} = \sum_{b,u} r_{b,u,t,\tilde{p}}\, R_{b,u,t} \quad \forall \tilde{p}, t,$$

$$S_t = \sum_{b,u} s_{b,u,t}\, R_{b,u,t} \quad \forall t,$$

$$E_{CO2terr,t} = S_t - S_{t-1} \quad \forall t,$$

and for climatic state (in vector form):

$$\Gamma_t = \phi \cdot \Gamma_{t-1} + \lambda \cdot E_t \quad \forall t,$$

where:

- $\beta$ is the (periodic) discount rate,
- indices $t, p, c$ and $e$ refer to time period, process, commodity and emission type,
- $K_{p,t}, C_{p,t}$ and $A_{p,t}$ are the investment to, capacity and activity of process $p$ at time $t$,
- $E_{e,t}$ is the emission of type $e$ at time $t$,
- $c^K_{p,t}$ and $c^A_{p,t}$ are the investment and operation costs of process $p$ at time $t$,
- $c^E_{e,t}$ is the emission penalty of emission type $e$ at time $t$ (if applicable),
- $I_{p,c,t}$ and $O_{p,c,t}$ are input and output flows of commodity $c$ to or from process $p$,
- $i_{p,c,t}$ and $o_{p,c,t}$ are the ratios of commodity $c$ input and output to process $p$ activity,
- $d_{c,t}$ is the end-use demand for the commodity $c$ at time $t$ ($d_{c,t} = 0$ if not applicable for commodity $c$),
- $f_{p,t}$ is the average capacity/availability factor process $p$ at time $t$,
- $\tau_p$ is the lifetime of process $p$, and
- $e_{e,c,t}$ is the emission factor for emission type $e$ and process $p$ at time $t$,
- indices $b$ and $u$ refer to biome and land-use type,
- $R_{b,u,t}$ is the land-area in biome $b$ for land-use $u$ at time $t$,

- $R_b$ is the total land-area of biome $b$,
- $\tilde{p}$ refers to processes producing a single land-use commodity,
- $r_{b,u,t,\tilde{p}}$ is the yield of the commodity produced by $\tilde{p}$ in biome $b$ for land-use $u$ at time $t$,
- $S_t$ is the terrestrial carbon stock at time $t$,
- $s_{b,u,t}$ is the terrestrial carbon density per area in biome $b$ for land-use $u$ at time $t$,
- $\Gamma_t$ is a vector of climatic state variables at time $t$,
- $\phi$ is the state-transition matrix of the climatic state,
- $\lambda$ is a transfer matrix from the emission vector to the climatic state, and
- $E_t$ is a vector of all emission types at time $t$.

The objective function is thus to minimize the discounted costs from investments and the operation of the processes. When applying emission pricing, which can be differentiated by emission category as needed, the objective function also accounts for the cost of emissions and thus incentivizes emission reductions. The first set of constraints portray commodity production and use in the energy and materials systems. These require that all commodity balances are satisfied (production is greater than use, including externally specified demand), process activity is constrained by capacity, and capacity results from investments and retirements (i.e., capacity has a fixed lifespan). Last, GHG emissions arise from processes' activities according to the processes' emission factors.

The land-use equations presented in Equation (1) are a considerably simplified version the actual set of equations in SuCCESs, i.e. its dedicated land-use model CLASH (Ekholm et al., 2024). This simplified set of equations, nevertheless, gives an overview of how land-use is modelled: land-area is divided into different biomes, each biome's area can be distributed between different land-uses. Each biome and land-use type can yield certain land-use commodities, and their vegetation and soil contain a carbon stock. The net $CO_2$ flux from the atmosphere to terrestrial biosphere is calculated as the difference of the carbon stock over consecutive time periods. This simplified representation does not portray the dynamics for forest growth or soil carbon stocks, however. A full depiction is given in the model description of CLASH (Ekholm et al., 2024). As CLASH is fully integrated into SuCCESs, all land-use decisions and their interaction with the other parts of the model are considered in the optimization problem portrayed in a simplified form by Equation (1).

For climate, Equation (1) provides only an abstract portrayal in matrix form, where all climate variables, i.e., atmospheric concentrations, radiative forcing and mean temperature increase, are aggregated into a single state vector $\Gamma_t$. A more detailed description is given in section 2.4.

Model users can additionally introduce new case-specific constraints that the model solution needs to satisfy; for example, a maximum limit for global mean temperature increase for investigating the cost-effective strategies to reach a specified temperature target. The following subsections describe each module of SuCCESs in more detail.

## 2.1 Energy

The production and use of energy and materials are modelled using the OSeMOSYS framework (Howells et al., 2011). This part of the model is very similar to well-known IAMs such as MESSAGE (Huppmann et al., 2019) or TIAM (Loulou and Labriet, 2008). It portrays the production and use of numerous energy and material commodities, and the capacities and operation of processes that convert the commodities from one form to another. The processes are described through techno-economic parameters, including input-output conversion ratios, capital costs, variable costs, capacity factors, and emission coefficients; compiled from various sources, such as the IEA Energy Technology Perspectives reports (International Energy Agency, 2020a) and numerous industry-specific reports.

The processes and commodities form a production network. For energy, this starts from the extraction of primary energy, passes through different transformation processes, and finally leads to the end-use of energy or energy services, depending on the sector. The model is driven by specified future demand projections for the end-uses, which it needs to satisfy. SuCCESs has been calibrated to IEA and other industry statistics for the base year 2020, and currently includes future energy and material demands following the SSP2 scenario until 2100 (Riahi et al., 2017). An aggregated overview of the production network is portrayed in Figure 2.

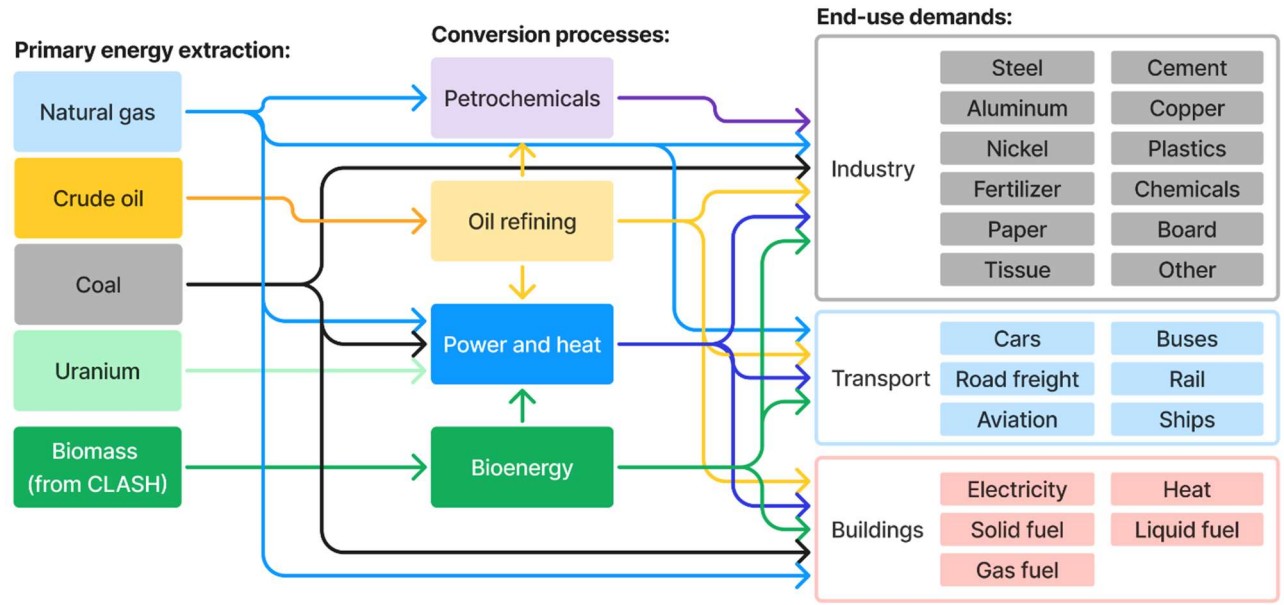

**Figure 2: A simplified overview of the energy system structure in SuCCESs. Hydro, wind and solar power are not depicted explicitly in the figure, as they consume no energy inputs, but are contained within the Power and heat sector.**

### 2.1.1   Primary energy extraction

Primary energy extraction covers coal, crude oil, natural gas and uranium. These are represented with the estimated available resources and their extraction costs for different regions, differentiating between proven and unproven resources according to International Energy Agency (2020b) estimates. Estimates for the average cost of extracting fossil fuels (in \$/GJ) was collected for the 23 largest producing countries, and these estimates were assumed to be representative for respective region.

### 2.1.2   Refining

The refining sector converts crude oil, natural gas and natural gas liquids into end-use fossil fuels and feedstocks for material production. Oil refineries process crude oil to a mix of liquified petroleum gases, gasoline, light oil (includes diesel and kerosene) and heavy oil. The input-output ratios of these four commodities are determined as a convex combination of four operating modes, ranging from low conversion (higher output of heavy fractions) to high conversion (higher output of lighter fractions) with reduced overall input-output efficiency due to more energy-intensive conversion. In addition to conventional oil refineries, SuCCESs includes the 'Crude oil to chemicals' (COTC) concept (Corma et al., 2017; Zhou et al., 2022), which produces higher shares of feedstocks for materials production than conventional refineries. These can be operated with similar convex combination of three operating modes, ranging from higher shares of light feedstock (ethylene) to mid-weight feedstock (propylene and C4 olefins) and finally to BTX (benzene, toluene and xylenes). Further conversion of fossil fuels to feedstocks can be done through steam cracking and fluid catalytic cracking, providing the model with more flexibility to meet the demand of each feedstock. The refining sector also includes the electrolysis-based hydrogen production, and the conversion of solid biomass into bioliquids.

### 2.1.3   Electricity and heat

Power plants in SuCCESs produce electricity and heat for low-temperature (heat) and high-temperature (steam) uses. Power plants are represented primarily through their conversion efficiency from fuel to end-product. Availability factors account for downtime, which affects the capacity to meet constant demand level. Efficiency factors and capital costs, including their assumed development into the future, are primarily based on Krey et al. (2019), although slightly adjusted for 2020 to better match with IEA statistics (Energy Statistics Data Browser) and interpolated for 2030. Electricity transmission losses are assumed to be 8%, based on IEA statistics.

The share of wind and solar has risen rapidly in electricity generation globally during recent years. The rising market share and the temporal variation of their production leads to the 'cannibalization effect', which depresses their revenues and thus limits these technologies' competitive market potential (e.g. Reichenberg et al., 2023). This can be mitigated through electricity storage (e.g. Ekholm and Virasjoki, 2020). Capturing variability in and between wind and solar production and electricity demand variation requires a temporally explicit model with sufficient resolution to capture the interplay of these variations. Further, modelling electricity storages requires representing chronology of consecutive hours over a sufficient

timeframe, e.g. for several days. As chronology is not considered in the time-slice-based approach of e.g. OSeMOSYS and TIMES (Howells et al., 2011; Loulou and Labriet, 2008) or a peak-load factor (Huppmann et al., 2019), we implemented a separate, hourly sub-module for electricity.

The electricity sub-module code and data are based on Ekholm and Virasjoki (2020), representing the hourly-level variations of wind and solar power capacity factors and electricity demand over four representative weeks, covering combinations where the average capacity factors over the week of both production types are either high or low. Variation in demand is represented in relation to the average demand. Hourly energy balances require that electricity demand equals to the sum of dispatchable generation, variable renewable generation and electricity storage net-discharge in each hour. Dynamic equations between consecutive hours account for the dynamics of electricity storages, looping from the last to the first hour of each week to avoid end-of-horizon effects. Each week has a weight for the share of a year that it represents. To bridge between the hourly sub-module and the main SuCCESs energy module, the weighted sum of the weeks' production and consumption needs to be equal to the annual balances. The production capacity for each production process is likewise obtained from the main SuCCESs energy module, as well as the changes in technologies' average capacity factors over the decades.

Some care is required to interpret the representation of the hourly electricity sub-module properly. Given that SuCCESs is a global, single-region model, the accurate representation of the electricity system is not possible. The real-world consists of geographically fragmented market areas with constraints for transmission between them, the average capacity factors of wind and solar and their variability differ between locations, as well as the variability of demand. This cannot be captured in a single-region representation. Yet, the model aims to represent the global phenomenon of variability, and the SuCCESs electricity sub-module tries to capture and represent this in a physically explicit manner. Additionally, the chronological hourly implementation allows the explicit modelling of electricity storages, which are likely to become an important part of the electricity system following the rapid increase in wind and solar power, as discussed above.

### 2.1.4 Industry

The representation of industry in SuCCESs consists of two approaches: explicit modelling of selected bulk materials' production, and aggregated modelling for the rest of industrial energy use. The explicit part includes specific production processes to produce steel, concrete, and plastics, for example, and is presented in more detail in section 2.3. For the aggregated industrial production classified as 'Other industry', energy use equals the difference between the modelled explicit processes and the statistics for industrial energy use (Energy Statistics Data Browser). For this part, energy demand is considered separately between solid, liquid and gaseous fuels, electricity and heat, with perfect substitution allowed between fossil and biogenic fuels.

### 2.1.5 Transportation

The transportation sector in SuCCESs considers various transportation modes separately: passenger cars, buses, road freight, rail passengers and freight, domestic and international aviation, and maritime freight. Demand for each mode is expressed as annual passenger-kilometres for passenger transportation, or as annual tonne-kilometres for freight transport. The projections are based on the ITF Transport Outlook 2021 (OECD/ITF, 2021) until 2050 and extended to 2100 following the SSP2 scenario (Riahi et al., 2017). Passenger cars are modelled explicitly through the number of cars, differentiating between gasoline, diesel, hybrid and electric vehicles. Due to lack of detailed data and the diversity of the vehicle fleet, other transportation modes are considered more abstractly through the average energy use to produce a unit of end-use demand (passenger-km or tonne-km), accounting for different energy sources.

### 2.1.6 Buildings

The buildings sector is the least detailed of the energy end-use sectors in the current version of SuCCESs. The sector aggregates residential and commercial buildings and adds the other energy-use categories from International Energy Agency (2024) not included in industry or transportation. As with the 'Other industry', the building sector considers directly the solid, liquid and gaseous fuels, electricity and heat consumption of the aggregate sectors, with future projections based on the SSP2 scenario (Riahi et al., 2017). While substitution between fossil and biofuels is allowed, no explicit modelling of the energy services is included in the current version of the model.

### 2.2 Land-use

The biophysical aspects of land-use are represented with a dedicated model, CLASH (Ekholm et al., 2024), which is hard-linked with the rest of the SuCCESs IAM. CLASH models the allocation of land area to different uses, including production in agriculture and forestry, terrestrial carbon stocks in vegetation and soil, and $CH_4$ and $N_2O$ emissions from land-use activities. Global land area is divided into ten biomes to capture the vastly different conditions for vegetation growth and carbon stock dynamics around the world. The land area in each biome can be allocated to different land-uses, including agriculture, forestry, and primary ecosystems. The biophysical properties of land and vegetation – such as vegetation carbon density, crop yields and forest growth – are biome-specific and respond to climate change. CLASH is parametrized to emulate the dynamic global vegetation model Lund-Potsdam-Jena General Ecosystem Simulator (LPJ-GUESS) (Lindeskog et al., 2021; Smith et al., 2001, 2014) in different future climate scenarios. An overview of CLASH is presented in Figure 3. A full description is provided in (Ekholm et al., 2024).

The energy and materials modules are linked with CLASH through the production of different types of biomass. As other parts of SuCCESs consider the world as a single-region, the commodity production and GHG emissions from CLASH are summed from the biomes to the global level in the linking between the modules. Crops satisfy the demand for food crops and livestock feed, but can be used as bioenergy, e.g. as feedstock to crop-to-ethanol processes. Wood is produced as three

commodities: logs satisfy the material wood demand, pulpwood is used for pulping processes to meet paper and board demand, and other wood can be used for energy. The harvest of crops and trees also generates agricultural and forestry residues, which can be either collected for energy use, incurring costs and a loss in the terrestrial carbon stock, as the harvest residues would otherwise be deposited on the litter carbon stock. The linkage with the climate module occurs through the $CH_4$ and $N_2O$ emissions of agriculture, and through the atmosphere-land carbon flux, which is calculated as the change in terrestrial carbon stocks over time.

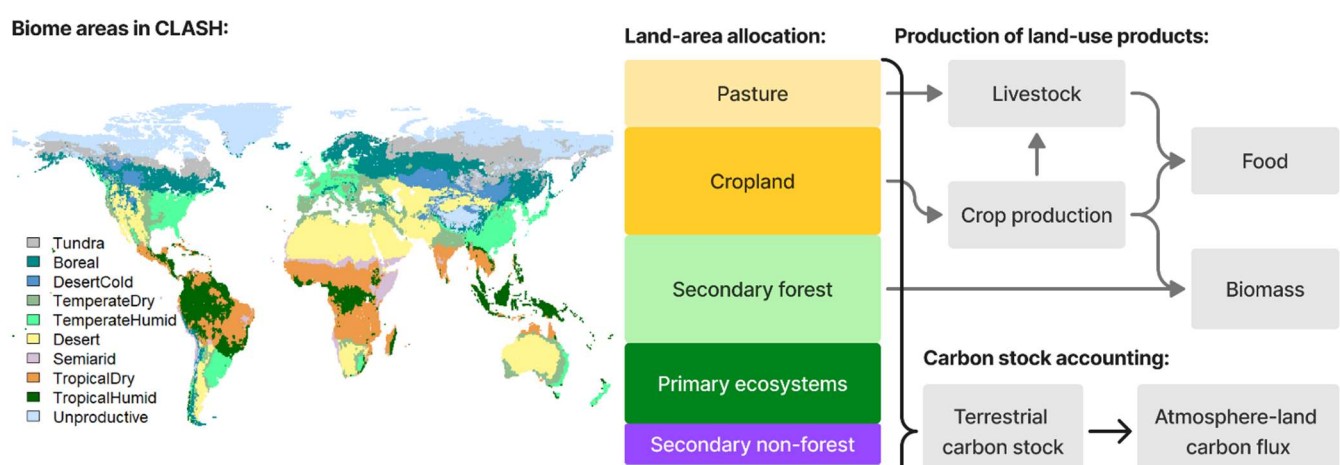

**Figure 3: An illustration of the main components in CLASH. Global land-area is divided into 10 biomes, each having different climatic conditions and thus vegetation growth, crop yields and terrestrial carbon stock dynamics. The area in each biome can be allocated to different uses. Cropland is used to produce crops for human consumption, livestock feed and biomass. Secondary forests are modelled by age cohorts. Harvesting forests produces timber, pulp wood and energy wood. Terrestrial carbon stocks cover vegetation, litter and soil. Vegetation carbon stocks are calculated from the area and biome and land-use specific carbon densities. Vegetation produces litter, which partly decomposes into the atmosphere and is partly deposited on soil. Soil carbon stocks decompose slowly and release carbon dioxide into the atmosphere. The atmosphere-land carbon flux is calculated as the change in terrestrial carbon stocks over time.**

## 2.3 Materials

In the current model version, the material system covers explicitly the production of main bulk materials, capturing their energy and fossil and biogenic raw material needs, and process emissions. The covered materials are steel, concrete, aluminium, copper, zinc, paper, board, tissue, wood logs and plastics. These were selected due to the high the energy-use, emission and land-use impacts of their production (see e.g. International Energy Agency, 2020a). Demand projections are specified for each material. Ongoing model development is aimed at extending this by including the dynamics of in-use stocks of materials, including stock decay, waste generation and recycling. A longer-term aim is to integrate material and energy systems more closely, so that the capacity expansion and retirement are accounted for within the material stocks themselves.

Steel production is represented with four different processes: blast furnaces (with and without carbon capture and storage, CCS), direct-reduced iron with electric arc furnaces, and recycled steel using electric arc furnaces. Aluminium, copper and nickel production cover similarly both the production of virgin material and the use of scrap metal. The scrap production routes require significantly less energy but are constrained by the availability projections of scrap materials in the current implementation.

The pulp and paper industry covers the production of three pulp types: chemical pulp, mechanical pulp and recycled pulp; and three end-products: paper, board and tissue. Chemical and mechanical pulp require pulpwood as input, although in different quantities, and both processes have significantly different electricity and steam requirements. While recycled pulp requires much less energy input, it is constrained by the availability of recycled paper. The three end-products require the three pulp types in different ratios, with paper being mostly produced from chemical pulp, board from a mix of three, and tissue using much more recycled pulp than the paper and board.

Plastics are divided into two categories: thermoplasts, which are easily recyclable, and thermosets, which are not. Both are produced from a fixed mixture of six constituent feedstocks: ethylene, propylene, C4+ olefins, BTX, ammonia and methanol; in ratios based on Levi and Cullen (2018). Feedstock production is covered in the refining sector, as described in Section 2.1.2. These feedstocks also serve as inputs for solvents and other petrochemical products, and ammonia for nitrogen fertilizer production.

## 2.4   Climate

Climate change is modelled with a simplified representation similar to the climate module of the TIAM model (Syri et al., 2008). Emissions of $CO_2$, $CH_4$ and $N_2O$ increase the gases' atmospheric concentrations, with $CH_4$ and $N_2O$ undergoing first-order decay processes with atmospheric lifetimes of 9.1 and 131 years, respectively (IPCC, 2013). Natural $CH_4$ and $N_2O$ emissions are from Saunois et al. (2020) and Tian et al. (2024), respectively. A four-reservoir model is used for $CO_2$. Differing from other similar representations of atmospheric $CO_2$ dynamics (e.g. that in Syri et al., 2008, or Nordhaus 1993 and 2017), the representation in SuCCESs climate module covers only the carbon exchange between the atmosphere and oceans, as the atmosphere-land carbon flux is already covered by CLASH (Ekholm et al., 2024). The four-reservoir model was calibrated following two guiding principles. First, the steady-state net atmosphere-ocean $CO_2$ transfer profile was calibrated to match the mean of multiple Earth System Models (ESMs) compared in (Joos et al., 2013). Second, the model's remaining free parameters (the initial amount of carbon in the different ocean reservoirs) were calibrated so that the atmospheric $CO_2$ stock simulated with the model corresponds to those from the MAGICC model in three alternative emissions scenarios (Meinshausen et al., 2011). With this parametrization, the model produces a present-day net atmosphere-ocean $CO_2$ flux of 2.5 GtC per year, which is consistent with the 2.8±0.4 GtC range reported by Friedlingstein et al. (2023).

Atmospheric $CO_2$, $CH_4$ and $N_2O$ concentrations $P_e(t)$ affect radiative forcing $RF(t)$, which is modelled as a logarithmic function of $CO_2$ concentration, and a joint radiative forcing effect from $CH_4$ and $N_2O$ concentrations, based on (Ramaswamy

et al., 2001). Radiative forcing from other sources is given exogenously and added to the total radiative forcing. In the standard parametrization of SuCCESs, the exogenous RF equals the average of SSP2-2.6 scenarios, but this assumption can be adjusted to be compatible with the intended scenarios. The total radiative forcing $RF(t)$ is then the sum of the individual components:

$$RF(t) = RF_{CO2}(t) + RF_{CH4}(t) + RF_{N2O}(t) + RF_{exo}(t), \tag{2}$$

$$RF_{CO2}(t) = \alpha \ln\left(\frac{P_{CO2}(t)}{P_{CO2,0}}\right),$$

$$RF_{CH4}(t) = \beta\left(\sqrt{P_{CH4}(t)} - \sqrt{P_{CH4,0}}\right) - \left(f\left(P_{CH4}(t), P_{N2O,0}\right) - f\left(P_{CH4,0}, P_{N2O,0}\right)\right),$$

$$RF_{N2O}(t) = \gamma\left(\sqrt{P_{N2O}(t)} - \sqrt{P_{N2O,0}}\right) - \left(f\left(P_{CH4,0}, P_{N2O}(t)\right) - f\left(P_{CH4,0}, P_{N2O,0}\right)\right),$$

$$f(P_1, P_2) = 0.47\ln(1 + 2.01 \cdot 10^{-5}(P_1\,P_2)^{0.75} + 5.31 \cdot 10^{-1}\;P_1\,(P_1\,P_2)^{1.52}),$$

with the parameters $\alpha$, $\beta$ and $\gamma$ from (Ramaswamy et al., 2001). As the functions of radiative forcing are non-linear, they were linearized in the average concentration between 2020-2100 in SSP2-RCP2.6 scenario to maintain the linear formulation of SuCCESs.

The change in global mean temperature is calculated using a three-reservoir model adopted from the simple climate model FaIR (Leach et al., 2021). The FaIR implementation transforms the reservoirs into a three-component impulse-response representation, where the temperature change $\Delta T(t)$ is the sum of the components $S_i(t)$:

$$\Delta T(t) = \sum_{i \in \{1,2,3\}} S_i(t), \tag{3}$$

$$S_i(t) = q_i\,RF(t)\left(1 - e^{-\frac{\Delta t}{d_i}}\right) + S_i(t - \Delta t)e^{-\frac{\Delta t}{d_i}}$$

where $q_i$ and $d_i$ are respectively the default radiative response parameters and timescales of FaIRv2.0.0, and $\Delta t$ is the model time step, i.e. ten years.

## 3   Comparison between historical data, SuCCESs and SSP scenarios

As a demonstration of SuCCESs, we run a baseline scenario and three scenarios with end-of-century targets for radiative forcing and compare the model's main results to historical estimates and SSP2 scenarios with corresponding radiative forcings (Riahi et al., 2017; Rogelj et al., 2018). In this setting, SuCCESs finds the least-cost strategy to reach the specified radiative forcing level by 2100. To align more closely with the SSP scenarios, we constrain the areas of different land-uses in each biome to match the harmonized LUH2 land-use SSP2 scenario (Hurtt et al., 2020). No additional constraints were

imposed on the model, such as limiting the rate of technology deployment or annual investments. We present the comparison in two parts: Section 3.1 portrays the scenarios produced by SuCCESs under standard parametrization; while Section 3.2, presents a sensitivity analysis with a random sampling of the main input parameters, leading to a 1000 scenario sample for each radiative forcing target.

## 3.1    Scenarios from a standard SuCCESs parametrization

Figure 4 shows the main energy flows from the four SuCCESs scenarios, along with historical values of fossil and biomass primary energy supply, electricity use and variable renewable electricity (VRE) from International Energy Agency (2024). These are compared to the range of SSP2 scenarios. The SuCCESs results for the year 2020 closely match the IEA statistics with a minor exception in bioenergy, which results from that SuCCESs does not include traditional bioenergy use. The

365 volume of traditional bioenergy is small compared to commercial bioenergy and its volume is not expected to grow in the future, and hence this omission has minor implications for the scenarios calculated with SuCCESs.

When compared with the SSP scenarios, the most notable differences occur with the baseline scenario without a radiative forcing target. Whereas coal use increases steadily in the SSP baseline scenarios and VRE generation remains modest, the converse happens in the SuCCESs baseline. This reflects the rapid decline in wind and solar power costs during recent years,

which is accounted for in the SuCCESs technology parametrization, but assumably occurs only later in the SSP scenarios. Natural gas use, however, grows strongly in the SuCCESs baseline, as it offers a low-cost solution to cover the gap between the variable wind and solar generation and demand. Oil use remains approximately at current levels in the SuCCESs baseline, driven by efficiency improvements and electrification in transportation, whereas most SSP2 baseline scenarios project increasing oil use.

In the scenarios with radiative forcing targets, the SuCCESs scenarios exhibit many key features of the SSP2 mitigation scenarios: fossil fuel use declines, with coal having the most pronounced effect; while biomass use, electrification and VRE generation increase with more stricter climate targets. The SuCCESs scenarios do not exhibit a full phase-out of crude oil and natural gas, however, and biomass use remains lower than in the SSP2-1.9 scenarios. These stem from the absence of transportation biofuels in the SuCCESs solutions, as transportation energy demand is satisfied with electricity and some

residual fossil fuel use. Although the presented scenarios included land-use constraints that restrict maximum bioenergy potentials, we have made similar observations in mitigation scenarios with less constrained land-use. This suggests that SuCCESs prefers biosphere carbon sink enhancement over cropland expansion and biofuel production as a cost-effective mitigation measure.

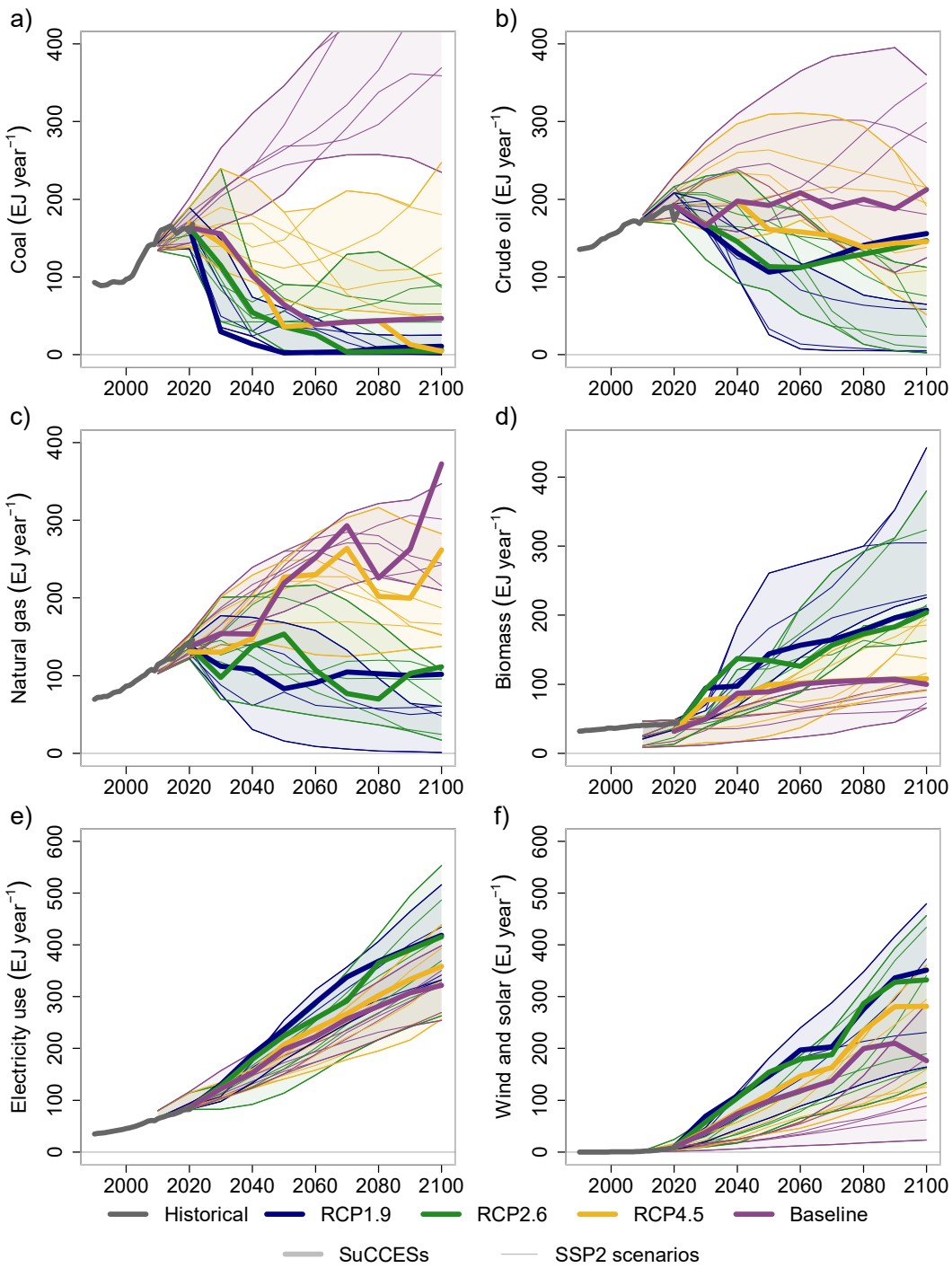

**Figure 4: Comparison of a) coal, b) crude oil, c) natural gas, d) biomass and e) electricity use, and f) electricity generation by wind and solar between historical values (grey), and baseline, RCP4.5, RCP2.6 and RCP1.9 scenarios (indicated by colour) from SuCCESs (thick lines) and other IAMs from the SSP database (thin lines, with shaded areas indicating the range).**

Figure 5 presents similar comparisons for $CO_2$ emissions from fossil fuels and industrial processes, net $CO_2$ emissions from land-use, and anthropogenic $CH_4$ and $N_2O$ emissions. For land-use emissions, we present the net $CO_2$ emissions from managed land and deforestation, rather than presenting total $CO_2$ flux between the atmosphere and terrestrial ecosystems. Historical estimates for these emissions are drawn from datasets by Smith et al. (2023) and Gütschow et al. (2016, 2024). The SuCCESs results align well with the historical estimates, particularly when considering the notable uncertainty in land-use $CO_2$ emissions and anthropogenic $CH_4$ and $N_2O$ emissions.

When comparing to the SSP2 scenarios, the SuCCESs baseline has far lower fossil $CO_2$ emissions, in line with the lower coal us in the SuCCESs baseline presented in Figure 4a. In scenarios with a radiative forcing limit, however, SuCCESs results align well with the range of their SSP2 counterparts. Net $CO_2$ emissions from land-use exhibit a declining trend in line with both the historical estimates and SSP2 scenarios, with stricter radiative forcing targets leading to negative land-use emissions by mid-century. Baseline $CH_4$ emissions remain lower than in the SSP2 scenarios particularly due to a lower projected headcount of ruminant livestock. Similarly, baseline $N_2O$ emissions remain lower than in the SSP2 scenarios, primarily due to the more modest growth of cropland $N_2O$ emissions. The current implementation for cropland $N_2O$ emissions in SuCCESs is to apply a constant emission factor for the cropland area, which doesn't account for e.g. changes in nitrogen fertilizer application over time. One direction for future model development is a better integration of nitrogen fertilizer use, crop yields, and cropland $N_2O$ emissions, which is likely to correct this difference between scenarios. CCS deployment, presented in Figure 5e, starts slightly later and remains cumulatively lower than in the SSP2-1.9 and SSP2-2.6 ensemble, and is non-existent with the RCP4.5 target. Carbon prices also remain considerably lower than in the SSP scenarios, which is attributable to the competitiveness of wind and solar power in the SuCCESs scenarios in relation to the SSP ensemble, as was discussed in relation to Figure 4.

The main climate variables are presented in Figure 6. Atmospheric $CO_2$ concentrations, and align well with historical estimates and SSP2 scenarios, apart from the baseline, for which SuCCESs exhibits significantly lower emissions and thus concentrations than the set of SSP2 baselines. Radiative forcing from the endogenously modelled $CO_2$, $CH_4$ and $N_2O$ concentrations also align rather well with both historical estimates and the SSP2 scenarios, although the linearizations used in SuCCESs introduce a small error with the lowest and highest levels of radiative forcing. For global mean temperature increase, the impulse-response representation from FaIR used in SuCCESs reacts slightly differently than the MAGICC model used in the SSP projections (Leach et al., 2021), but the projected temperature changes remain similar overall.

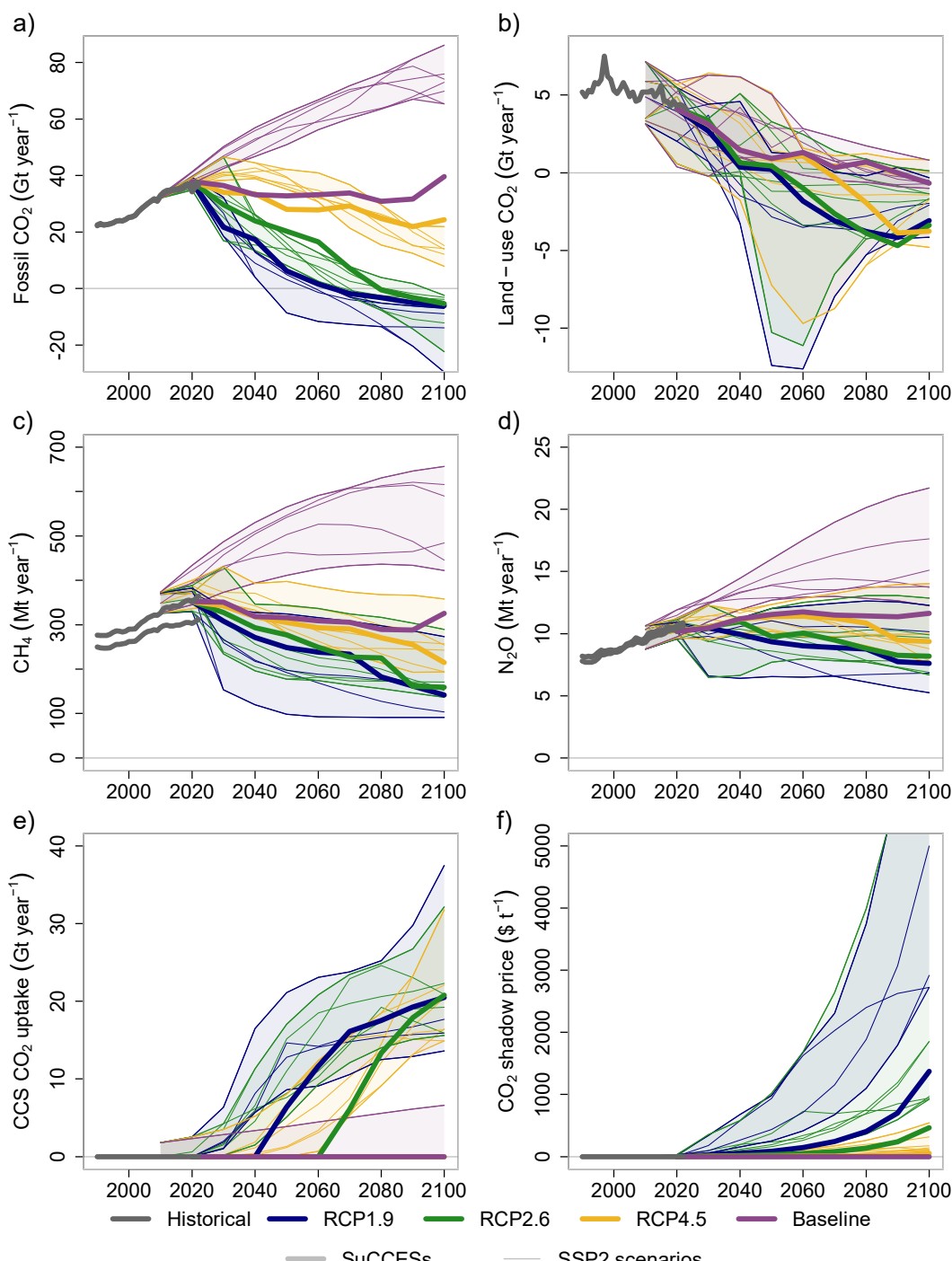

**Figure 5: Comparison of a) fossil and industry CO₂, b) land-use CO₂, c) CH₄, and d) N₂O emissions, and e) CO₂ uptake by CCS and f) carbon prices between historical estimates (grey), and baseline, RCP4.5, RCP2.6 and RCP1.9 scenarios (indicated by colour) from SuCCESs (thick lines) and other IAMs from the SSP database (thin lines, with shaded areas indicating the range).**

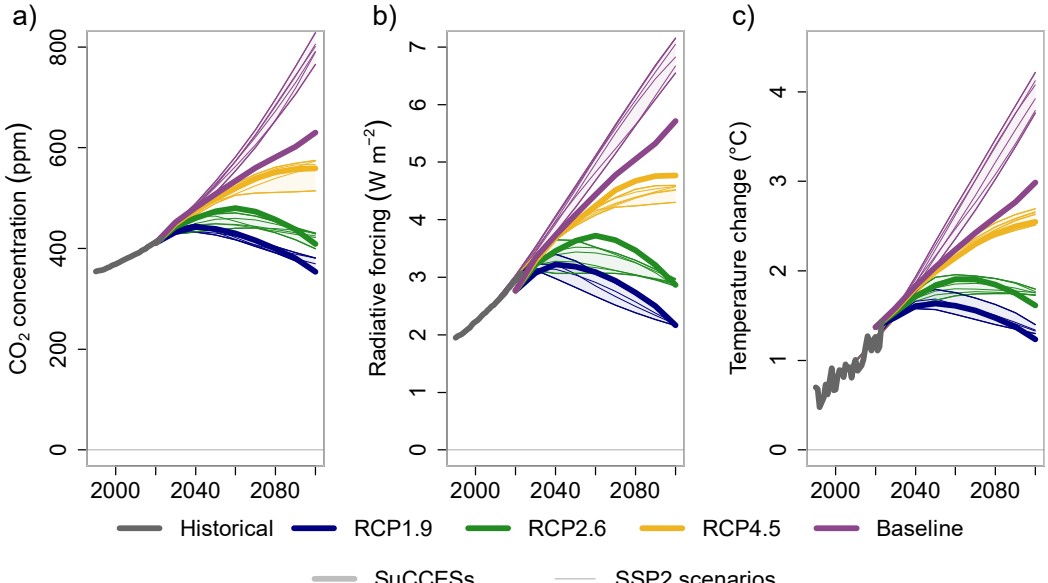

**Figure 6: Comparison of a) atmospheric CO₂ concentration b) radiative forcing by CO₂, CH₄ and N₂O; and c) global mean temperature increase between historical estimates (grey), and baseline, RCP4.5, RCP2.6 and RCP1.9 scenarios (indicated by colour) from SuCCESs (thick lines) and other IAMs from the SSP database (thin lines, with shaded areas indicating the range).**

## 3.2 Sensitivity analysis

The purpose of IAMs is to explore a range of potential futures, and each resulting scenario is always a function of the model structure and input data. Therefore, the scenarios presented above in Section 3.1 are only a small subset of possible futures that SuCCESs can produce. Given that SuCCESs is computationally lightweight[2], it can easily run a large set of scenarios, allowing for a broad scenario exploration across a wide range of alternative futures.

To demonstrate this, and to assess how sensitive SuCCESs is to changes in input assumptions, we conducted a simple sensitivity analysis where the main input parameters are varied around their default values using Monte Carlo sampling, after which the model is solved for the radiative forcing targets as in Section 3.1. We are aware of a single study (Panos et al., 2023) where a process-based, bottom-up IAM has been used with Monte Carlo sampling, using a sample size of 1000 for four scenario families. Our approach is similar in that we also run 1000 scenarios for each radiative forcing target. However, the implementations differ slightly. The aim of Panos et al. (2023) was to quantify future uncertainty resulting from parameter uncertainty, therefore using realistic probability distributions for the input parameters to represent this uncertainty. As our aim is to investigate the sensitivity of SuCCESs to changes in input parameters, we vary each parameter within a uniformly sampled +/-20% range around its default value. For production technologies, the varied parameters include variable and capital costs, input-activity ratios, capacity factors, lifetimes, and start years. For land-use we varied the

---

[2] A single run with the current model version takes approximately 10 seconds with a standard laptop using CPLEX solver.

vegetation carbon densities, crop yields and forest fire probabilities; and for model-wide parameters the discount rate, projected commodity and service demands, and the amount of fossil fuel resources. No correlations are set between different parameters' variations. The +/-20% range was chosen, as it spans small variations that are plausible (e.g. the capital cost of wind power in 2050 is 960 $/kW instead of 1200 $/kW), but which can still be large-enough to affect the results. The variation of future demands required us to relax the land-use constraints, whereby the lower bound of cropland, pasture and secondary forest areas in each biome were set to 90% of the LUH2 SSP2 scenario values.

Similar to the results in Section 3.1, the results for main energy flows are presented in Figure 7 and GHG emissions in Figure 8. Even relatively small changes in the input parameters induce large changes in fossil fuel use in the baseline scenario, with the SuCCESs scenario ensemble spanning a wider range than the SSP2 scenarios used as a reference here. Natural gas use increases strongly in many scenarios without a radiative forcing target, and these scenarios also exhibit high electricity consumption and low generation with coal and wind. In scenarios with stricter radiative forcing targets, the range of fossil fuel use is more constrained. The range of electricity use and VRE generation in the SuCCESs ensemble spans most of the range that the SSP2 ensemble covers. However, the scenario ensemble members exhibit bioenergy use that reaches only the lower range of SSP2-1.9 scenarios in 2100. Nevertheless, electricity use, VRE and bioenergy are more prominent in scenarios with more strict radiative forcing targets.

Similarly, $CO_2$ emissions from fossil fuel and industry (Figure 8) vary widely in the baseline scenario ensemble, but are effectively constrained by the radiative forcing target. Net $CO_2$ emissions from managed land show less variations across the different radiative forcing cases, but lower radiative forcing targets consistently lead to more negative emissions during the latter half of the century. Some 'spikes' appear in certain periods, resulting from forest clearing in that period, and these occur jointly with the increase of biomass use observed in Figure 7 panel d). The behaviour of $CH_4$ and $N_2O$ emissions is more stable, responding predictably to the radiative forcing targets.

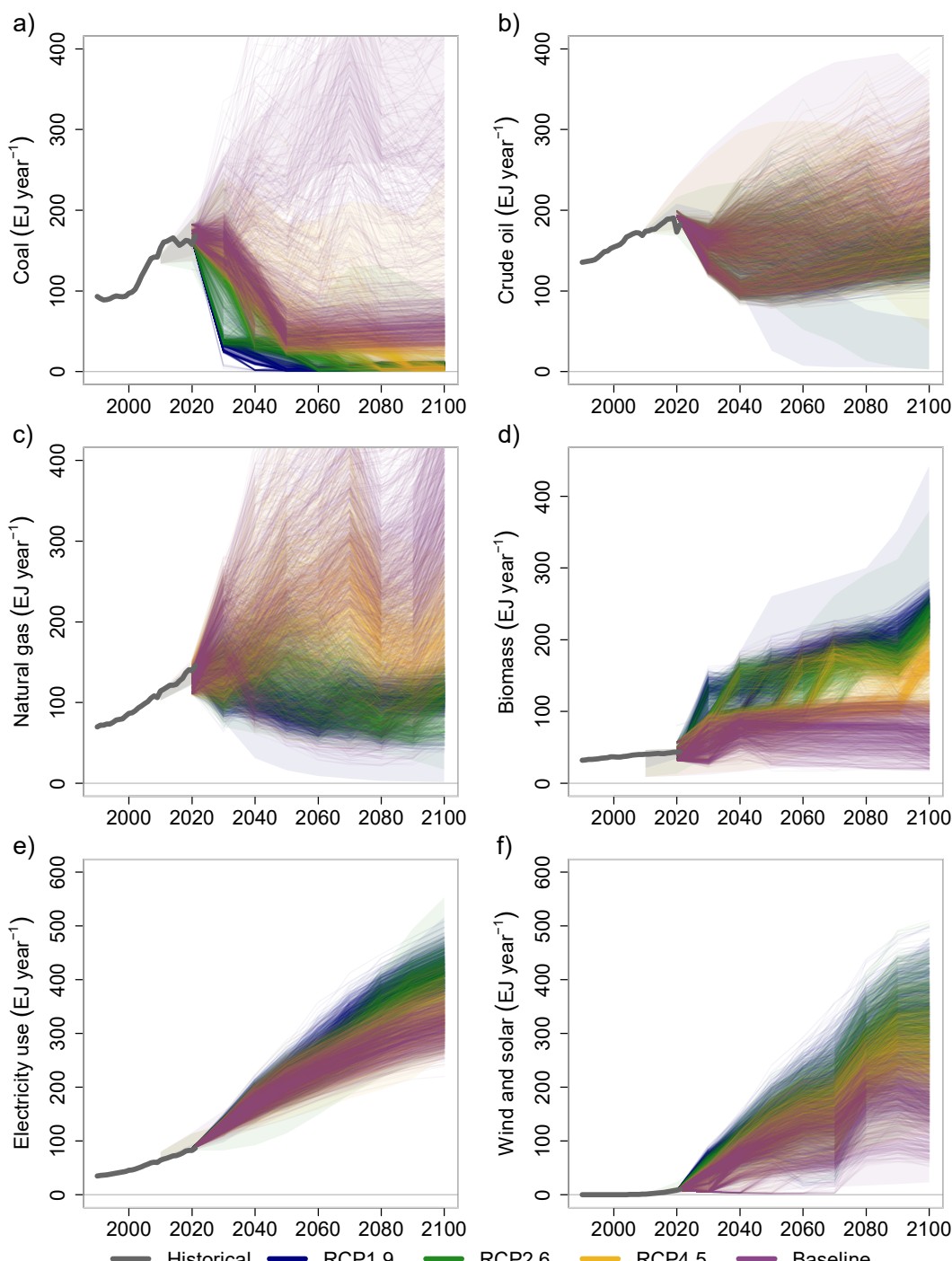

**Figure 7: Comparison of a) coal, b) crude oil, c) natural gas, d) biomass and e) electricity use, and f) electricity generation by wind and solar between historical values (grey), and baseline, RCP4.5, RCP2.6 and RCP1.9 scenarios (indicated by colour) from SuCCESs (lines) and the range from other IAMs' results in the SSP database (shaded areas).**

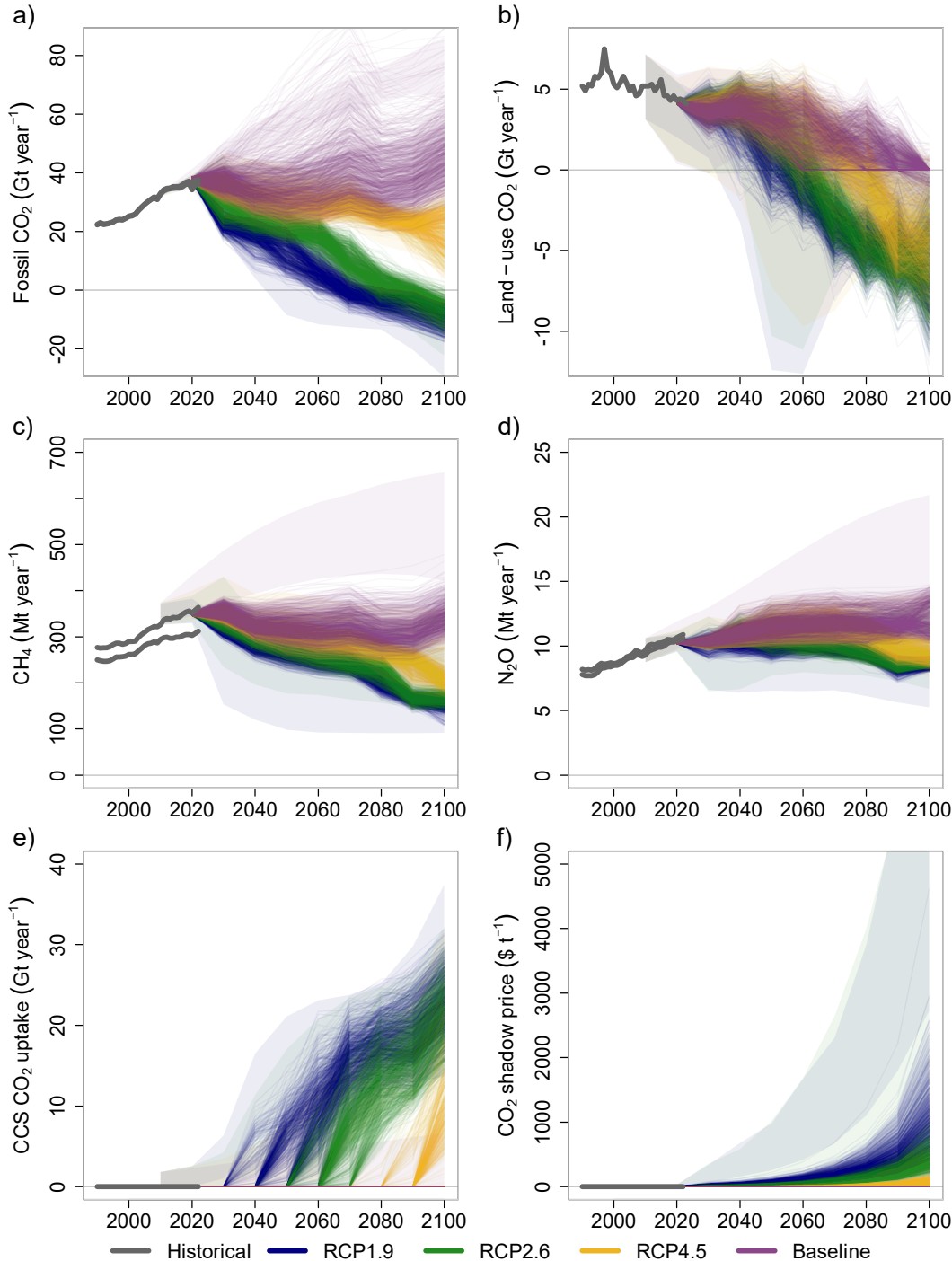

**Figure 8: Comparison of a) fossil and industry CO₂, b) land-use CO₂, c) CH₄, and d) N₂O emissions, and e) CO₂ uptake by CCS and f) carbon prices between historical estimates (grey), and baseline, RCP4.5, RCP2.6 and RCP1.9 scenarios (indicated by colour) from SuCCESs (lines) and the range from other IAMs' results in the SSP database (shaded areas).**

## 4    Using the model

The SuCCESs IAM is openly available on GitHub[3], where further practical instructions on using the model can be found. To run the model, General Algebraic Modeling System (GAMS) and a suitable optimization solver are required. Although a linear programming (LP) solver is preferred to take advantage of the model's linear formulation, we have also run the model successfully with nonlinear (NLP) solvers. Straight out of the box, the model produces results that closely align with the

baseline scenario presented in Section 3.1. SuCCESs has a minimal number of 'custom' constraints that steer the scenarios toward specific behaviour. Instead, modellers can introduce additional constraints related to land-use, climate variables, technological deployment, and other factors as needed in each modelling experiment. Alternative scenario specification can be created by varying the input parameters, as demonstrated through Monte Carlo sampling in Section 3.2. The main file, 'SuCCESs.gms', includes a dedicated section where scenario files containing additional constraints or parameter variations

can be introduced.

To assist new users in exploring the SuCCESs model output, a Python toolbox is also available for download on GitHub[4]. The toolbox includes instructions for getting started, a list of the essential output variables with descriptions, and ready-to-use python code for data processing and plotting. The README file within the GitHub repository provides guidance on setting up the toolbox, including troubleshooting tips for installing the necessary gdx-pandas package, which is required for

reading GAMS model output in the GDX format.

The toolbox is organized into multiple python files categorized by topic. We strongly recommend that new users closely follow the import function of the toolbox (*import_gdx.py*), which cleans and filters the data in various ways and filters out the essential output variables (*essential_outputs.txt*). Other files contain basic calculations and aggregations, such as converting GHG emissions to $CO_2$-equivalents. The basic plotting functions within the toolbox serve three primary purposes:

first, to demonstrate the various types of outputs; second, to provide a starting library for users' own analyses; and third, to offer a quick control tool for verifying the plausibility of model runs. It is important to note that the plotting functions are not exhaustive but serve as a foundation for further customization. Finally, the toolbox features a Jupyter notebook tutorial (*tutorial.ipynb*) that showcases all provided functions and plots, facilitating an intuitive learning experience for new users.

## 5    Conclusions

SuCCESs is a bottom-up, global IAM with hard-linked energy, materials, land-use and climate modules. Its key novelty is in the hard-linked integration of energy, material, land-use and climate systems that are solved through intertemporal optimization (Keppo et al., 2021). This allows to calculate long-term cost-optimal scenarios within the considered domains without the computationally heavy iteration between soft-linked models. While most bottom-up IAMs operate with 11 to 61

---

[3] https://github.com/SuCCESsIAM/SuCCESsIAM
[4] https://github.com/SuCCESsIAM/SuCCESsIAM-toolbox

regions (Keppo et al., 2021), SuCCESs adopts a more aggregated approach by representing the world as a single region, but preserving some regional detail in land-use through the 10 climate zones of CLASH. This simplified geographic resolution – together with the linear programming implementation of the intertemporal optimization problem – makes SuCCESs computationally lightweight: a single scenario run takes roughly 10 seconds on a standard laptop using the CPLEX solver. This comes with the obvious trade-off that SuCCESs cannot provide results on a regional level, and the low geographic resolution can introduce biases in results that depend more strongly on regional characteristics or trade between regions. Together, the features of SuCCESs render it particularly suited to address questions where the interactions between the systems (hard-linking) and over time (intertemporal optimization) are vital, and explored through large scenario ensembles (low computational burden).

When running scenarios with radiative forcing targets for year 2100 (van Vuuren et al., 2011), SuCCESs produced results that closely match historical estimates of energy use and GHG emissions, and align well with the range of scenarios observed in the SSP scenario ensemble (Riahi et al., 2017). However, SuCCESs also differs from the SSP scenario ensemble in some respects: SuCCESs scales down coal power and uses more wind and solar energy already in its baseline, relies less on bioenergy for mitigation, and does not phase-out crude oil or natural gas fully even under the ambitious $1.9\,Wm^{-2}$ radiative forcing target.

Yet, the sensitivity analysis of Section 3.2 revealed that some of these observations are features of the default model parameterization. Even small to moderate variations in input parameters (up to +/-20%) can result in a diverse set of scenarios, spanning in some cases a broader range than those observed in the SSP scenario ensemble. This suggests that there are multiple ways to satisfy the projected future demands and climate targets that are nearly cost-optimal with the default model parametrization, so that small to moderate variations in the model inputs can lead to rather different optima (Neumann and Brown, 2023). This diversity is also reminiscent of the fact that a model is not fixated to a set of results, but merely translates inputs into outcomes, and model results are inherently dependent on the underlying input data and assumptions. This observed diversity in the resulting scenarios also aligns with one of the main purposes of IAMs and scenario models in general: to explore a broad range of potential futures and illuminate the long-term consequences of different policy and technology pathways (Trutnevyte et al., 2016).

SuCCESs' computational efficiency makes it particularly well-suited for large-scale scenario explorations. For instance, as demonstrated in Section 3.2, it can easily produce large scenario ensembles e.g. through Monte Carlo sampling – an approach rarely applied in bottom-up IAM. We know so far only one study where a bottom-up IAM has been run with Monte Carlo sampling (Panos et al., 2023). Looking ahead, the model could be extended to a stochastic programming framework, allowing it to endogenously acknowledge the uncertainty and learning on selected input parameters and find optimal hedging strategies to mitigate risks, e.g. in climate sensitivity and future technologies  (e.g. as in Schaber et al., 2024). We know a few such experiments carried out with bottom-up IAMs, although with relatively limited scenario trees focusing on a single source of uncertainty (Labriet et al., 2012; Loulou and Kanudia, 1999; Syri et al., 2008).

This paper documents the initial version of SuCCESs and some early scenario experiments. Current model development focuses on expanding the material module to include in-use stocks of materials and creating a regionalized version of the model with more detailed representation of the electricity system. Additional to broadening the model scope and improving its representation of the world, we are working on *how* the model is used, e.g. doing large-scale scenario exploration and analysing the preconditions for selected future events. We believe that broadening the existing set of models, models' scope and innovative modelling approaches can yield new insights into diverse pathways the world could take towards a sustainable future. This can lead to even more valuable tools for exploring such possibilities and guiding decision-makers through complex questions regarding sustainability and climate policy.

*Code availability*. The current version of SuCCES is available in GitHub: https://github.com/SuCCESsIAM/SuCCESsIAM under the MIT License. We have archived on Zenodo the version used in this paper (DOI: 10.5281/zenodo.13981520), as well as the results and scripts used to plot the results in Section 3 (DOI: 10.5281/zenodo.13981206).

*Author contributions*. TE formed the initial idea and structure of the model. Model development was carried out by all authors. TE carried out the model analysis for this manuscript, with contributions from other authors. All authors contributed to writing the paper.

*Competing interests*. The authors declare that they have no conflict of interest.

*Acknowledgements*. The research has been done with funding from the Research Council of Finland in projects SuCCESs (decision number 341311), OptiMit (decision number 331491) and RealSolar (decision number 358543).

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
