# Peer review of "SuCCESs – a global IAM for exploring the interactions between energy, materials, land-use and climate systems in long-term scenarios (model version 2024-10-23)"

_Geoscientific Model Development, 2024_

## Author Response (AR1)

**Response to Reviewers (GMD-2024-196)**

We would like to thank all three reviewers for their insightful and constructive comments, which were very helpful in refining our paper. Below, we provide a response to each comment. The submission of the revised manuscript includes a version with the changes indicated with red text color.

**Reviewer #1:**

The study introduces SuCCESs, a bottom-up Integrated Assessment Model (IAM) that integrates energy, materials, land-use, and climate systems globally to explore long-term scenarios through 2100. By hard-linking these systems, SuCCESs captures interactions such as greenhouse gas emissions and their climate impacts. Scenarios are solved using intertemporal optimization, minimizing system costs while meeting demand and climate constraints, yielding equilibrium solutions. The article details the model structure, evaluates its performance against other IAMs under varying radiative forcing targets, and demonstrates its capacity for large-scale scenario exploration through a Monte Carlo sensitivity analysis with a 1000-member ensemble. Below are specific comments and questions to enhance clarity and address potential uncertainties.

Below are specific comments and questions to enhance clarity and address potential uncertainties:

Major:

1. The climate system in SuCCESs endogenously includes only three GHG emissions ($CO_2$, $CH_4$, and $N_2O$). How do exogenous inputs for radiative forcing from other sources influence the outcomes of mitigation scenarios? Could the authors discuss the potential uncertainties introduced by these exogenous factors?

   - Yes, we now elaborate this in section 2.4 of the revised manuscript. The exogenous radiative forcing is directly added to the calculated RF from $CO_2$, $CH_4$ and $N_2O$. Being exogenously defined, the exogenous RF is not affected by the outcome of the optimization procedure. To ensure consistency, one can adjust the exogenous RF trajectory to be compatible with the intended scenarios.

2. Given the varying levels of detail in the representation of the energy, land, material, and climate systems, could the authors provide guidance on scenarios or cases where SuCCESs excels (e.g., scenarios benefiting from its lightweight design and detailed system representation)? In which scenarios or cases might the model produce higher uncertainties, and why? Such guidance would be valuable for potential users of the model.

- Thank you, this is a good point. We have expanded the discussion regarding these aspects in sections 1 and 5, also addressing e.g. the first comment by Reviewer #2. For example, in the Conclusions we now summarize this as follows: "Together, these features of SuCCESs render it particularly suited to address questions where the interactions between the systems (hard-linking) and over time (intertemporal optimization) are vital, and explored through large scenario ensembles (low computational burden)."

3. Lines 373–374: "Even small to moderate variations in input parameters (up to ±20%) can result in a diverse set of scenarios, spanning in some cases a broader range than those observed in the SSP scenario ensemble." How does this model's sensitivity to input parameters compare with other IAMs? Is this sensitivity within a reasonable range?

- We are not aware of similar sensitivity analyses done with other bottom-up IAMs that would allow a comparison, as the Monte Carlo experiment by Panos et al. was also a bit different in its scope. Instead, we have extended the discussion of this topic in the revised manuscript. An interpretation for this diversity in the sampled scenarios is that there are multiple ways to produce energy and reduce emissions that are nearly cost-effective. Then, should technologies develop one way or another (reflected in the variations of the technology parameters of SuCCESs), one technology might be adopted while another wouldn't, leading to potentially very different scenarios.
- Note: the cited lines were 473–474 (not 373–374) in original submitted version, i.e. the Conclusions section, and we have modified that part accordingly.

4. The calibration, extensions, and exogenous inputs mentioned in the article are based on SSP2, e.g. in Lines 164, 233, 243, and 313. Additionally, the comparison with other IAMs and the sensitivity analysis are also based on SSP2. Does the model already include other SSPs (i.e., SSP1, SSP3, SSP4, and SSP5)? If so, and if it is not too much effort, how does the model perform under these SSPs compared to other IAMs? Providing this information would be helpful for potential users. However, I understand that comparing SuCCESs' performance under other SSPs with other IAMs may require significant effort. Even just clarifying whether the model includes other SSPs in the article would still be very useful.

- At the moment, we have SuCCESs parametrization only for SSP2, which we now state more clearly in section 2.1 of the revised manuscript. We are working on the other SSP implementations, but this will take some time as it's part of a larger effort.

Minor:

1. Line 74, does the assumption imply that all demands are fixed? What are the potential impacts of this assumption on the model results, especially in scenarios where demand might vary dynamically?

- We have now clarified this statement. Demand changes over time (i.e. it is a 'projection') but is fixed (i.e. inelastic) are each model period.

2. Line 74, how does the model account for supply elasticity? Please clarify how flexible responses in supply are modeled across different systems.

- We have now clarified this statement. The supply-side comprises many technologies and resources with different costs, which together constitute the supply-side elasticity.

3. Figure 2, why are wind and solar energy sources not included in the figure?

- The reason is that they do not require any energy inputs that need to be extracted. Hydropower is not depicted in Figure 2 either, for the same reason. As this conforms with the representation of these production technologies in SuCCESs (i.e., wind, solar and hydro processes have no inputs, but only an output), we chose not to present them in the figure, but mention in the figure caption that they are contained within the "Power and heat" box.

4. Line 344: "SuCCESs does not include traditional bioenergy use." How might the exclusion of traditional bioenergy affect future projections of energy use and related emissions? Please discuss the implications.

- We now discuss this further. The volume of traditional bioenergy is rather small compared to commercial bioenergy (as was visible in Figure 4) and its volume is not expected to grow in the future, and hence this omission has minor implications for the scenarios calculated with SuCCESs

5. Line 334, how is land use constrained in the model? Additional details on the mechanisms or assumptions for land-use constraints would be useful.

- For these scenarios, we fixed the land areas per use and biome to those of the LUH2 SSP scenarios, which is now stated more clearly. Essentially, any land-use constraints are determined by the user and can be configured to the study setting as desired, as discussed in section 4.

6. Does the energy and material system provide all three major greenhouse gases (CO2, CH4, and N2O) to the climate system? If so, how are these emissions generated by the energy sector? Additional details on this linkage would be helpful.

- We now clarify this at the beginning of section 2, i.e. that all three modelled GHG emissions arise from energy production and use, material production and land-use. Energy-related emissions arise from combusting the carbon embedded in fossil fuel, as well as volatile CH4 emissions and N2O from combustion; while certain industrial processes emit also non-energy-related process emissions.

7. Figures 5, 6, and 7, the shaded areas representing the range of results from other IAMs in the SSP database are not easy to distinguish. Could the authors consider revising the visualization methods to make these areas clearer for readers?

- Yes, we have adjusted the visualizations to improve readability. We increased the opacity of the shaded areas and strengthened their outlines, so that the shaded areas stand out better; and also made the individual lines narrower, so they produce less clutter that impedes readability. This was not done for the Monte Carlo results, however, as the SuCCESs scenarios are overlaid with the shaded areas, and the individual SSP scenarios are not presented in those figures.

**Reviewer #2:**

The study introduces SuCCESs, a lightweight global Integrated Assessment Model (IAM), providing documentation and test results. Overall, the paper is well-written and well-structured, offering comprehensive technical details of the modeling framework. While there are already several widely used IAMs, I believe there is value in developing new models. However, I encourage the authors to elaborate on the unique aspects of SuCCESs, add more comprehensive results, and discuss the limitations and future directions. Below are my detailed comments:

- We're glad to hear that the Reviewer found the paper well-written and structured, and the model development itself valuable. We appreciate the comments that encourage elaborating the description, results and discussion in the revised manuscript.

Motivation:

The introduction effectively summarizes the background of IAMs and categorizes SuCCESs as a bottom-up, process-based model relative to existing literature. While the information is

useful, the research or modeling gap justifying the need for this new model remains unclear. Apart from its "lightweight" computational feature, which offers advantages in computational efficiency at the expense of resolution, the rationale for developing SuCCESs is insufficiently discussed. The inclusion of a comparison with SSP scenarios is appreciated; however, reproducing existing results is not necessarily a novel contribution. I recommend the authors emphasize the specific gaps SuCCESs addresses to ensure it makes a meaningful impact.

- Thank you, we have now expanded on the rationale for developing the model in the introduction and conclusions. There are two main rationales: being lightweight, as mentioned also above; and providing a hard-linked combination of bottom-up energy, materials, land-use and climate modules that are solved through intertemporal optimization. We now summarize this in the following way: "Together, these features of SuCCESs render it particularly suited to address questions where the interactions between the systems (hard-linking) and over time (intertemporal optimization) are vital, and explored through large scenario ensembles (low computational burden)."
- Comparison with the SSP scenarios is not a novel scientific contribution, but it is provided as an "evaluation against standard benchmarks, observations, and/or other model output", as requested in the guidelines for model description papers published in GMD.

Furthermore, SSP scenarios are somewhat outdated. The authors should consider incorporating results from the IPCC AR6 scenario database or the recent SSP v3 database (IIASA), which includes updated GDP and population projections and is being evaluated under ScenarioMIP.

- Indeed, the SSP scenarios are somewhat outdated, and we also note this in the manuscript. The AR6 database includes a broader set of more recent scenarios, but also these are not very recent (publication dates range from 2012 to 2021). Additionally, due to their diversity in underlying scenario assumptions, these are not as usable as a reference point for model comparison. On the other hand, the recent SSP v3 database contains so far (as of February 2025) only GDP and population projections, as the reviewer notes, and not the energy and emissions pathways that we would need for the comparison.

Model Features and Design:

SuCCESs is presented as a single-region model, but the associated CLASH model seems to offer more regional granularity. Could the authors clarify how SuCCESs and CLASH are coupled, given their differing regional resolutions?

- Yes, this is now explained better in the revised manuscript in sections 2 and 2.2. Basically, the agricultural and forestry outputs, GHG emissions etc. from CLASH are summed to arrive at the global quantities, which are used on the other modules of SuCCESs.

CLASH appears to have a detailed representation of agricultural and land-use sectors. Does it include price-responsive food demand and international trade?

- CLASH is purely biophysical model, which we now express more clearly in section 2.2, so it does not cover demand or trade. Additionally, all demands in SuCCESs are inelastic and trade is not considered explicitly (implicitly assuming perfect markets for all products on the global level). These are now expressed more clearly in the revised manuscript, in the beginning of section 2. However, users can assign additional constraints that e.g. require a certain level of self-sufficiency in food production based on the population in each CLASH biome, as we are doing in one forthcoming manuscript.

While a single-region model offers computational simplicity, it assumes complete global market integration, ignoring regional market differences. Recent studies, such as Zhao et al. (2022; DOI: 10.1016/j.gloenvcha.2021.102413), underscore the importance of international trade in IAMs. Discussing the trade-offs involved in this design choice would be valuable.

- Indeed. Trade-offs like these determine to what questions a model is suited to address and how it performs. These are now discussed more extensively in section 5 of the revised manuscript.

Calibration and Baseline:

Around line 163, the paper mentions that future energy and material demands are calibrated to SSP2 until 2100. Does this imply that demand is fully exogenous? Similarly, the assumptions regarding supply and demand (e.g., whether they are exogenous or endogenous) in the context of partial economic equilibrium modeling should be clarified. For instance, around line 173, the description of resource extraction costs suggests these are used to construct supply curves. More clarity here would be beneficial.

- Yes, the demand projections are exogenously determined and demand is inflexible. The supply side is elastic, due to the varying costs of alternative technologies and resources, as the reviewer also discusses above. We have now clarified these aspects at the beginning of section 2, as suggested.

The Baseline scenario shows emissions lower than those in the SSP database. Could the authors elaborate on whether this discrepancy is an artifact of model assumptions? If the definition of the baseline differs from SSP baselines, the comparison might not be consistent.

- Yes, the SuCCESs baseline (i.e. scenario with no constraint on 2100 radiative forcing) has notably lower emissions than the corresponding SSP scenarios. This result stems from the high deployment of wind and solar energy and decline of coal use, which occurs also in the baseline with SuCCESs. As this happens with no climate policy, the definition of baseline is consistent with the SSPs. This explanation was given only in relation to Figure 4, we now discuss this more explicitly also in relation to Figure 5.

The constraint on land-use to LUH2 (Hurtt et al., 2020) is mentioned. What would the implications be if this constraint were not applied? Additionally, could the model incorporate land mitigation policies, such as differentiated carbon prices across scenarios?

- If one runs SuCCESs without any constraints on land-use, CLASH would 'shuffle' land-use across biomes extensively, relocating croplands, pastures and managed forests to areas based on their relative competitive advantages in that particular scenario. As this is not very realistic from the standpoint of self-sufficiency, for example, we typically constrain land-use in some way. For the comparison with the SSPs, constraining land-use to LUH2 made the scenarios most comparable, and hence we applied that constraint in the scenarios of this paper.
- SuCCESs can indeed incorporate land mitigation policies, either through a constraint (e.g. on temperature increase or RF, as was done in the scenarios of section 3), whereby SuCCESs seeks mitigation measures equally from energy, material production and land-use; or through emission pricing, which can be differentiated across emission categories (e.g. by fossil/land-use emissions, or by GHG). In both cases, SuCCESs considers also the dynamics and interactions between these systems. We mention these aspects now in section 2 of the revised manuscript.

Results:

Including a broader range of results, such as carbon prices, climate outcomes, and final energy and agricultural demands, would strengthen the paper.

- Indeed. We have now extended the results to cover additional variables, e.g. carbon prices in Figures 5 and 8 (previously Fig. 7) and the main climate variables in new Figure 6.

Minor Comments:

Figure 6: Were the axes truncated inappropriately?

- No, but on purpose, to keep them consistent with each other, with Figure 4, and to keep the low-end of y-axis more readable. Should we extend the y-axis so that all the extreme realizations of the baseline would be visible, this would make the scenarios with RF targets hard to read.

Line 370: Clarify the reference to "IPCC."

- On line 370, there is a reference to IGCC. We now clarified the reference.

Model Uniqueness: Will the model solution always be unique?

- It is not guaranteed. There can be multiple minima that have the exact same costs. For example, if there are two ways to produce a given resource with the exact same extraction costs, then it does not make a difference for the model which way the resource is extracted. The practical relevance of this is minor, as other variables of the model would not be altered between these different minimizing solutions.

**Reviewer #3:**

The paper introduces the SuCCESs model, a new integrated assessment model (IAM) representing global energy, materials, land-use and climate systems for long-term scenarios exploration up to 2100. The model is designed to be highly agregated in one region, as to lower computational requirements and to allow for Monte Carlo simulations. This provides an interesting model application, that is not yet commoly seen in the field of IAMs.

The paper is easy to read and provides a comprehensive overview of the model structure, and its potential for application. However, I believe it misses a few important references to existing literature, and could be improved in its structure to better highlight the model key features and relevance for IAMs-related research.

- We're glad if the Reviewer found the paper easy to read and to provide a comprehensive overview of the model. We appreciate all the comments and suggestions, which enable us to improve the manuscript further.

Below, there are specific comments and questions that can be relevant to enhance the clarity of the paper and the contribution that SuCCESs can bring to field of research.

Specific comments:

Lines 37-41: it would be valuable to mention some specific examples of bottom-up process-based IAMs as described in this paragraph, with related references to the literature.

- Indeed. We combined this with the following comment, and refer to MESSAGEix and GLUCOSE, as they share the same solution concept and a partly same structure as SuCCESs.

Lines 46-48: I believe it would be worth mentioning here other IAMs similar to SuCCESs either in their systems representation and /or in their structural definition, and to clearly highlight how SuCCESs differ from them and what benefits it provides. Some examples: the GLUCOSE model, it provides a highly aggregated IAM developed using OSeMOSYS and representing energy, land and food, and materials systems (for reference, please see Beltramo et al, 2021 <https://doi.org/10.1016/j.envsoft.2021.105091>); the MESSAGEiX model, as you have briefly mention already it has recently started expanding its systems representation by adding a representation of materials flows and stocks (for reference, see Ünlü et al, 2024 <https://doi.org/10.5194/egusphere-2023-3035>).

- Yes, as stated in the previous response, we now cite GLUCOSE and MESSAGEix in the referred paragraph in the revised manuscript.

Line 85: could you please explain the reason behind the choice of a more detailed land-use representation as compared to the other systems in SuCCESs, and what are the benefits of this for the model application.

- Certainly. The reason is that vegetation growth, agricultural productivity and terrestrial carbon stock dynamics are very different e.g. in the Amazon area, Sahara, or Northern Europe. We now mention this more explicitly in section 2.2. Pooling all global land area would thus give very misleading results. The case is not as drastic with energy technologies and industrial processes: an oil refinery, for example, will work more or less in the same way in all of these locations. Energy resources do differ across locations, but also these are routinely shipped across the globe. The aspect of omitting trade (due to the model being single-region) is now mentioned in sections 2 and 5.

Line 86-87: could you please explain the reasons behind the choice of adopting a ten-year time-step model resolution, and what could be the benefits and drawbacks of this choice particularly in consideration of the fast pace at which the technology transitions are expected to take place in order to mitigate climate change? In addition, could you please clarify if the ten-year time steps are solved consequentially (i.e. assuming myopia) or all at once (i.e. assuming perfect foresight)?

- Yes. This was chosen in consideration of the trade-off between computational complexity and temporal resolution. Although technological transitions can be fast, modeling the intermediate steps (e.g. annual, every 5 years) was not considered to be critical enough to warrant the higher computational burden. The model is solved assuming perfect foresight, as was noted in the beginning of section 2.

Could you please clarify which parts of the SuCCESs model are individual, separate models that are hard linked to the SuCCESs model, and which parts instead are embedded elements of the SuCCESs model as it is? E.g. you mentioned in the text that the land-use system is represented using the dedicated CLASH model and the energy system is represented using the OSeMOSYS model: is there another dedicated model also used for the material sector?

- Yes, this is now clarified in section 2 of the revised manuscript. Basically, SuCCESs is a single model that is solved in one go, which is portrayed in equation (1). We developed and published the land-use part of the model separately, and it can be also used stand-alone, hence earning its own name: CLASH. OSeMOSYS is not exactly a model, but a modelling framework, which we use for the energy and materials production parts of the model, as stated in the beginning of section 2.1.

Lines 276-277: could you please provide a reference to the statement saying that the covered materials "were selected due to the high energy-use, emissions, and land-use impacts of their production"?

- It is somewhat difficult to find a single, definitive source for this kind of general knowledge, but we added a reference to an IEA Energy Technology Perspectives report, which discusses industrial energy use and emissions from these industries.

Section 4: I believe this section is not fitting in its current position towards the end of the paper, as it does not add to the results and the scientific contribution that the SuCCESs model provides, nor discuss further some of the model results presented in section 3. I would recommend the authors to either integrate some of the information provided here in section 2, under the model structure, or to simply remove this section and use it for a separate model documentation in the supplementary material or on the dedicated GitHub repository.

- We agree that the section does not add to the results or scientific contribution. Yet, it is stated in GDM policies that model description papers published in GMD should contain a user manual (with 'should' indicating that it is not obligatory but recommended). This short section is not a full-fledged user manual (which we think is

not practical to be published in a paper) but, nevertheless, gives potential users some guidance on what is needed and how to get started with using the model.

Line 461: I would recommend to move the reference to Keppo et al. (2021) if relevant to the earlier section of the paper (i.e. lines 86-87), also to better address comment 4 above.

- The referenced lines 86-87 of the initial submission and the comment 4 above refer to the temporal resolution of SuCCESs (10 year time steps). This topic is mentioned very briefly by Keppo et al. (2021), namely that timesteps in IAMs "usually vary from 1 to 10 years", which we think that doesn't bring much additional knowledge to warrant a citation in this context. However, the Keppo et al. (2021) paper is referenced already twice in the introduction, on lines 28 and 45.

Line 465: here it is mentioned that in SuCCESs "a single scenario run takes roughly 10 seconds on a standard laptop using the CPLEX solver". Considering such a fast solving time, could you please expand earlier on in the paper on the reasons behind choosing a 10-year time steps for the inter-temporal optimisation of the SuCCESs model?

- Indeed. The decision was made on the consideration of computational time vs. temporal resolution, as stated earlier. Of course, when making that decision (early in the model development process), we did not know what the solving time would be with a finished model. Now, however, that the run time is so short, it enables us to do e.g. Monte Carlo experiments such as those in section 3.2, or later to implement stochastic optimization capabilities, as discussed in the introduction and later in the conclusions.

Technical corrections:

Line 461: I would recommend the authors to check the wording of this sentence.

- Thank you, we have now clarified this sentence.

---

## Author Response (AR2)

**Response to Reviewers (GMD-2024-196, revision 1)**

We would like to thank the Referees for their helpful comments. We are happy that Referees #1 and #2 were satisfied with our previous responses; and Referee #3 for suggesting acceptance subject to minor revisions. We have now addressed the outstanding comments.

**Reviewer #3:**

General comments:

Thanks for the good answers to my previous review comments.

The revised manuscript has clearly been improved. However, I was still not able to find similar details and explanations – as provided in the answers to my review comments – reflected in the revised text. All my previous comments were expecting a related edit to the main manuscript. So, I would appreciate if you could simply include the answers you provided me in the revised manuscript accordingly.

- Thank you for the further feedback, and our apologies for not incorporating some of the information provided in the response letter to the manuscript. We have now done so, as indicated in the responses below.

Below, there are specific comments related to the information and clarification that I still perceive missing in the current revised manuscript.

Specific comments:

1. In lines 48-50 you wrote: "SuCCESs also differs from these models in terms of its single-region representation, which was chosen to maintain simplicity and low computational burden". This is incorrect, as the GLUCOSE model also provide one single region representation. I believe the key differences with the GLUCOSE model are in the detailed representation of the land sector and potentially in the materials representation. Please, check the literature and revise the text in the manuscript accordingly.

- Indeed. This statement was a remnant from the first version of the manuscript, which did not reference GLUCOSE. We have now modified the text to state "SuCCESs also differs from most of these models [...]".
- As the Reviewer notes, SuCCESs has a more detailed representation of land-use and materials production compared to GLUCOSE. However, we did not acknowledge GLUCOSE in a subsequent statement regarding recent expansion of model scope to material production. We have now added a citation to GLUCOSE there, as well.

[Authors' note: there was no Specific comment #2]

3. Line 95: could you please include in the manuscript a summary of the explanation you provided me as to why you chose to opt for a more detailed land-use representation as compared to the other systems, and what are the benefits of this for the model application? This information is still missing in the current revised manuscript.

- Yes, we now include a slightly shortened version of the response in the manuscript: "The reason behind the disaggregated representation for land-use is that vegetation growth, agricultural productivity and terrestrial carbon stock dynamics differ considerably around the Earth; while energy technologies and industrial processes function in more similar ways, and energy and other resources are routinely shipped across the globe."

4. Line 96-98: again, could you please include in the manuscript a summary of the explanation you provided me as to why you chose to adopt a ten-year time-step model resolution, and what could be the benefits and drawbacks of this choice particularly in consideration of the fast pace at which the technology transitions are expected to take place in order to mitigate climate change? Also, could you clearly add in the manuscript that you solved the model assuming perfect foresight?

- Indeed, the explanation is now given in the manuscript: "This was chosen in consideration of the trade-off between computational complexity and temporal resolution. Although technological transitions can be fast, modelling the intermediate steps (e.g. annual, every 5 years) was not considered to be critical enough to warrant the higher computational burden."
- Perfect foresight is mentioned in the first sentence of section 2, which introduces the overall model type: "SuCCESs is a global, demand-driven partial equilibrium model that is solved through intertemporal optimization (linear programming) assuming perfect foresight."

5. Section 4: I would still like for the authors to remove section 4 as it is and to either integrate some of the information in section 2, under the model structure, or to simply move this section in the appendix to the paper.

- We have now moved the section into the appendix, as suggested.